# 3D In Vitro Models: Novel Insights into Idiopathic Pulmonary Fibrosis Pathophysiology and Drug Screening

**DOI:** 10.3390/cells11091526

**Published:** 2022-05-02

**Authors:** Ana Ivonne Vazquez-Armendariz, Margarida Maria Barroso, Elie El Agha, Susanne Herold

**Affiliations:** Department of Medicine V, Internal Medicine, Infectious Diseases and Infection Control, Universities of Giessen and Marburg Lung Center (UGMLC), Cardio-Pulmonary Institute (CPI), Institute for Lung Health (ILH), Member of the German Center for Lung Research (DZL), Justus-Liebig University Giessen, 35392 Giessen, Hessen, Germany; margarida.barroso@innere.med.uni-giessen.de (M.M.B.); elie.el-agha@innere.med.uni-giessen.de (E.E.A.); susanne.herold@innere.med.uni-giessen.de (S.H.)

**Keywords:** 3D cultures, IPF modelling, drug screening

## Abstract

Idiopathic pulmonary fibrosis (IPF) is a progressive and often lethal interstitial lung disease of unknown aetiology. IPF is characterised by myofibroblast activation, tissue stiffening, and alveolar epithelium injury. As current IPF treatments fail to halt disease progression or induce regeneration, there is a pressing need for the development of novel therapeutic targets. In this regard, tri-dimensional (3D) models have rapidly emerged as powerful platforms for disease modelling, drug screening and discovery. In this review, we will touch on how 3D in vitro models such as hydrogels, precision-cut lung slices, and, more recently, lung organoids and lung-on-chip devices have been generated and/or modified to reveal distinct cellular and molecular signalling pathways activated during fibrotic processes. Markedly, we will address how these platforms could provide a better understanding of fibrosis pathophysiology and uncover effective treatment strategies for IPF patients.

## 1. Introduction

Idiopathic pulmonary fibrosis (IPF) is a chronic progressive interstitial lung disease associated with ageing and DNA damage that particularly affects adult males between 60 and 70 years old with a history of smoking [1]. While IPF aetiology remains unknown, environmental exposure to contaminants (e.g., dust or mould) and viral infections have been shown to increase the risk of contracting the disease [2]. Depending on the method used, IPF’s reported incidence and prevalence in Europe and North America varies between three and nine cases per 100,000 people per year [3]. Clinically, IPF is a devastating disease that displays a progressive lung function decline with a median survival of only 3 to 4 years [1]. Patients present persistent dyspnoea and dry cough while, at a cellular level, IPF is characterised by epithelial cell hyperplasia, alveolar consolidation, and myofibroblast activation [4,5].

Current diagnosis relies on typical radiology imaging with high-resolution computed tomography [6]. Until now, only nintedanib and pirfenidone, both anti-fibrotic medications, have shown promise in the amelioration of IPF progression [7]. Nintedanib is an intracellular suppressor of tyrosine kinases that acts as an inhibitor of fibroblast recruitment, proliferation, and differentiation, therefore negatively affecting extracellular matrix (ECM) deposition [8]. Alternatively, pirfenidone slows fibrosis generation by modulation of procollagen transcription and reduction of transforming growth factor (TGF)-β-induced fibroblast activation and differentiation [9].

Despite significant research being conducted, the majority of drug targets shown to attenuate induced fibrosis in pre-clinical models have failed in human clinical trial phases II or III [10]. A probable reason for these setbacks may be that animal models of IPF are generated artificially (e.g., application of bleomycin, silica and asbestos, age-related models, and cytokine overexpression) while the direct cause of the disease is still unknown [11]. Consequently, the identification of specific cellular and molecular mechanisms activated during IPF is essential for the understanding of crucial events that trigger unusual fibroblast outgrowth, progressive tissue scaring, and/or abnormal epithelial repair. Three-dimensional (3D) cultures have quickly become prominent tools for disease modelling, regenerative medicine, and drug development. Regarding the lung, hydrogels, precision cut lung slices (PCLS), lung organoids, and lung-on-chip have shown to be valuable techniques for the discovery and test of drugs in a variety of pulmonary diseases, including cystic fibrosis (CF), lung cancer, and chronic obstructive pulmonary disease [12,13,14].

In this review, we outline current 3D models used to elucidate cellular and molecular cues involved in IPF. We also discuss how each of these systems recapitulate specific elements of lung fibrosis and, to which extent, these systems and other cutting-edge in vitro technologies can be exploited to aid in the discovery of novel molecular targets for the treatment of IPF patients (Figure 1).

## 2. 3D Lung Culture Systems

3D culture systems display the cell-to-matrix and cell-to-cell communication lacking in common two-dimensional (2D) monolayer cultures. Mimicking the lung microenvironment in vitro enables 3D systems to assess cell structure and function during homeostatic and pathological conditions. In line with this, hydrogels are defined as water-swollen networks of polymers that allow cells to grow in a more physiological 3D shape. Hydrogels can be customised to better simulate the homeostatic or disease microenvironments by modifying the culture´s stiffness [15]. Another relevant in vitro model is the culture of PCLS obtained from ex vivo lung explants. PCLS contain nearly all cell types present in the lung, including epithelial cells, endothelial cells, smooth muscle cells, and fibroblasts. In culture, PCLS have a 3D configuration, remain metabolically active, and respond to cell-specific stimuli [16]. Alternatively, organoids derived from embryonic progenitors (ESC), induced pluripotent stem cells (iPSC), or adult tissue-resident stem cells have the capacity to proliferate and differentiate into 3D structures that resemble the organ of origin [17]. Notably, lung organoid systems have recently become valuable methods for investigating developmental organogenesis, disease, and regeneration [18,19]. Lastly, lung-on-chip systems introduce another level of complexity by implementing microfluidics. This technique recapitulates some of the organ functions by recreating lung, biochemical, and biomechanical microenvironments through continuous media perfusion and controlled mechanical stress [20]. In the next sections, we describe in more detail the applicability of each of these 3D lung in vitro systems for pulmonary fibrosis modelling and regenerative medicine (Table 1).

### 2.1. Hydrogels: Modelling of Human Lung Fibroblast Migration and Differentiation

Fibroblast foci and tissue stiffening are considered key pathological markers of IPF [4]. To implement more targeted studies, hydrogels with tunable matrix rigidities that allow fibroblast migration, myofibroblast differentiation, and ECM deposition have been employed to model how fibroblast phenotypes deviate in IPF patients. For example, cultures of human lung fibroblasts in stiff cultures promote upregulation of the transcription factor *megakaryoblastic leukaemia-1* (*MKL1*) and α-smooth muscle actin (αSMA) expression, resulting in myofibroblast differentiation and actin polymerisation. In this study, the mechanotransduction mechanisms that regulate myofibroblast differentiation were investigated by culturing fibroblasts in polyacrylamide (PA) hydrogels with different stiffnesses to mimic normal and fibrotic lung rigidities [21]. Notably, overexpression of nuclear *MKL1* was shown to promote fibroblast differentiation on soft matrixes, suggesting that actin polymerisation-dependent MKL1 nuclear accumulation is sufficient to override matrix stiffness-mediated cell differentiation. Furthermore, Asano et al. also showed higher expression of αSMA in primary human lung fibroblasts when PA hydrogels with higher stiffnesses were employed. In addition, fibroblasts stimulated with the chemotactic factor platelet-derived growth factor (PDGF) rapidly migrated and grew with dendritic extensions when cultured on stiffer gels. Interestingly, blockage of αSMA by short interfering RNA inhibited PDGF-mediated cell migration, indicating that αSMA is not only involved in human lung fibroblast differentiation but also regulates cell migration processes [22].

TGF-β is a potent pro-fibrotic cytokine secreted by myofibroblasts involved in the induction of lung fibrosis, particularly IPF [23]. In this regard, the relation between TGF-β and matrix stiffness was investigated by the culture of primary pulmonary fibroblasts isolated from healthy and IPF patients, in collagen-rich hydrogels with diverse rigidities and in the absence or presence of TGF-β. After culture of control and IPF fibroblasts in pro-fibrotic rigidities, TGF-β stimulation led to FAK/Akt signalling pathway activation, initiating collagen deposition by gene upregulation of *collagen type I alpha 1 chain* (*COL1A1*) along with inhibition of matrix metalloproteinase (*MMP*)-1 expression [24]. The authors’ results corroborate the role of the FAK/Akt pathway in collagen deposition in both normal and fibrotic environments. Accordingly, prostaglandin E2 (PGE2), a bioactive prostanoid, has been shown to prevent myofibroblast proliferation and differentiation by reducing the expression of *COL1A1* and αSMA in myofibroblasts [25]. PGE2 is synthetised from endogenous arachidonic acid through the cyclo-oxygenase (COX) pathway, particularly via COX2 after inflammatory stimuli such as TGF-β [26]. Remarkably, levels of both PGE2 and COX2 are considerably reduced in the lungs of patients with IPF compared to control individuals [27]. In concordance, PGE2 and COX2 expression in fibroblasts was also reduced when cultured in stiff matrixes, implying a direct correlation between matrix rigidity and PGE2-COX2 axis modulation [28]. In a recent study, terminal PGE2 synthetic enzyme prostaglandin E synthase (PTGES) was also reduced in the lungs of patients with IPF. Notably, human lung fibroblasts cultured in soft collagen hydrogels and spheroids showed a significant induction of the eicosanoid biosynthetic enzymes COX2, PTGES, and cytosolic phospholipase A2 compared to fibroblasts cultured in stiff plastic plates. These data indicate that lung stiffness in fibrotic regions may negatively affect the expression of multiple eicosanoid biosynthetic enzymes, halting PGE2 synthesis and, consequently, supporting fibrosis progression [29].

In another study, IPF-derived lung fibroblasts were shown to overexpress α6-integrin when cultured on stiff PA matrices. Data showed that upregulation of α6-integrin gene expression on stiff matrices was dependent on ROCK activation of the *c-Fos* and *c-Jun* transcription complex. Moreover, blockage of *c-Fos/c-Jun*-dependent α6-integrin-promoter activation by CRISPR interference technology or pharmacological inhibitors prevented stiff matrix-induced α6-integrin expression. Interestingly, along with other in vitro and in vivo models, the authors showed that α6-integrin mediates myofibroblasts invasion and lung fibrosis after injury, indicating that targeting this pathway might represent an attractive anti-fibrotic therapeutic strategy [30].

While hydrogels are not suitable in vitro models for the study of cellular interactions (e.g., epithelial-mesenchymal cell crosstalk), these studies demonstrate that the platforms are rather useful to recapitulate fibroblast cellular and molecular mechanisms occurring in vivo during fibrosis and are ideal for driving cell-specific therapeutic strategies.

### 2.2. PCLS: Uncovering Antifibrotic/Regenerative Pathways to Treat IPF

Besides myofibroblast activation and ECM deposition, it has been proposed that repetitive lung epithelial cell damage and reprogramming are closely involved in IPF pathogenesis [31]. In line with this, PCLS models are well-established methods that can be used to investigate the effect of potential drugs for the treatment of IPF patients [32]. For instance, Marudamuthu et al. used PCLS to demonstrate that a 7-mer deletion fragment of calveolin-1 scaffolding domain (CSP7) attenuates fibroblast activation. In vivo, CSP7 delivered systemically or locally improved the overall survival of bleomycin-injured mice by inhibiting not only the expression of ECM proteins but also by reducing the apoptosis of alveolar epithelial cells (AEC). Further, human end-stage IPF lung tissues treated with CSP7 showed a clear inhibition of the expression of pro-fibrotic proteins COL1A1 and α-SMA, supporting the potential use of CSP7 as a therapeutic target for IPF [33].

Given that IPF lung explants are rare and belong mostly to the end-stage of the disease, there was a need for human PCLS models that represent initial fibrosis to investigate early-stage pathomechanisms. In this regard, Alsafadi et al. used a cytokine cocktail containing TGF-β, PDGF-AB, tumour necrosis factor-α, and lysophosphatidic acid to create a model of early fibrosis using human PCLS derived from donors without IPF [34]. PCLS treatment with the fibrotic media upregulated the expression of pro-fibrotic (i.e., Actin alpha 2 (*ACTA2*), *MMP7*, and *Serpin family E member 1* (*SERPINE1*)) and pro-inflammatory markers (*Interleukin* (IL-)*1β* and *Wnt Family Member 5A*) without affecting cell viability. In addition, pro-fibrotic treatment-induced ECM deposition while reducing *surfactant protein C* (*SFTPC*) and *homeodomain-only protein homeobox* expression in AECII and AECI, respectively. In a follow-up study, 14 days after bleomycin or saline administration, PCLS obtained from injured mice was characterised by upregulation of mesenchymal fibrotic markers, fibronectin 1 (*Fn1*), and *Col1a1*, as well as increased secretion of total collagen and Wnt1-inducible signalling protein 1 (Wisp1). Moreover, treatment of control and bleomycin-treated PCLS with nintedanib showed increased *Sftpc* expression in fibrotic PCLS and attenuated *Wisp1* expression in both fibrotic and normal PCLS. Notably, pirfenidone treatment did not trigger *Sftpc* expression nor reduced Wisp1 secretion. For the human model, PCLS obtained from tumour-free lung tissue were treated with the mentioned pro-fibrotic cocktail to induce early fibrosis. Nintedanib treatment restored both *Sftpc* expression and SPC secretion, implying that recovery of AECII function might be a relevant feature of the anti-fibrotic mechanisms of nintedanib [35].

In the pursuit of novel molecular targets for IPF treatment, the role of senescence in lung epithelial cells after fibrosis was investigated by the measurement of SA-β-galactosidase activity, a surrogate marker for the detection of senescence cells, on PCLS obtained from PBS or bleomycin-challenged mice [36]. The authors showed a higher number of senescent AECs in fibrotic lungs, while treatment with senolytic drugs, dasatinib, and quercetin, resulted in reduced SA-β-galactosidase activity. In addition, treated AECs displayed a lower expression of the fibrotic markers, *Col1a1* and *Wisp1,* leading to the decline of the fibrotic burden and increased AECII marker expression [36]. These findings suggest a potential role of AEC senescence in the development of lung fibrosis and, if further confirmed in humans, identify senolytic drugs as possible candidates for IPF pre-clinical trials. Similarly, it was shown that metformin, an antidiabetic drug and an AMP-activated protein kinase (AMPK) activator, has a dual anti-fibrotic and lipogenic effect in murine and human models of IPF [37,38]. For instance, metformin was able to macroscopically attenuate fibrosis in human PCLS by relaxing the lung structure while increasing the number of lipid-containing cells and decreasing collagen deposition. These observations imply a transition from myofibroblast to lipofibroblast phenotype in metformin-treated PCLS. In combination with in vivo murine models, the authors showed that metformin activates AMPK signalling, which downregulates TGFβ-induced *COL1A1* expression. Along with AMPK upregulation, activation of a separate signalling pathway prompts lipogenic trans-differentiation of myofibroblasts by boosting bone morphogenetic protein 2 expression and peroxisome proliferator-activated receptor-gamma phosphorylation. These data highlight metformin as a promising and safe option for therapies that aim to accelerate fibrosis resolution.

In another recent study, the receptor-like protein tyrosine phosphatase eta (CD148/PTPRJ) mediated by its ligand syndecan-2 (SDC2) has been shown to also have anti-fibrotic effects in murine and human models of IPF. For instance, CD148-deficient mouse fibroblasts highly upregulated PI3K/Akt/mTOR signalling, suppressed autophagy, and promoted p62 expression, leading to nuclear factor kappa B activation and pro-fibrotic markers, *Fn1* and *Col1a1*, upregulation. Additionally, PCLS obtained from IPF and control individuals and treated with a pro-fibrotic mix presented an attenuated pro-fibrotic response when SDC2-based mimetic peptide was added to the media [39]. Furthermore, PCLS obtained from IPF patients, and bleomycin-injured mouse lungs treated with an inhibitor for the TGF-β regulators, αvβ6 and αvβ1 integrins, lead to lower *COL1A1* and *SERPINE1* expression [40]. Future studies will clarify if therapies focused on targeting CD148 ligands and/or αvβ integrins could become alternative clinical approaches to halt fibrosis progression in IPF patients.

Even though obtaining lung tissue is challenging, especially from patients, PCLS offers substantial advantages over traditional 2D cultures by preserving the lung architecture, harbouring multiple cell types, and being metabolically active in culture. Markedly, PCLSs support the identification of unknown molecular mechanisms underlying lung fibrosis and regeneration and aid the screening and discovery of anti-fibrotic/regenerative medications.

### 2.3. Lung Organoids: Mimicking of Pulmonary Fibrosis Pathophysiology for Personalised Medicine

Over the last decade, major progress has been made in organoid technology to develop structures that closely mimic the cellular and structural complexity of the in vivo scenario [41]. For instance, Sachs et al. generated human airway organoids (AO) from non-small cell lung cancer epithelial cells to create a model that could be applied to a variety of pulmonary diseases such as CF, lung cancer, and respiratory syncytial virus infection. AO contained polarised pseudostratified airway epithelium comprising basal, club, multi-ciliated, and secretory cells. Notably, passaged organoids retain similar cell frequencies independently of the number of passages, thus allowing long-term expansion. Regarding CF, AO lines obtained from the broncho-alveolar lavage of CF patients were characterised and proved to have a thicker layer of apical mucus, which was comparable to the in vivo phenotype [42]. As for translational approaches, Sette et al. described novel in vitro models, including organoids derived from primary airway epithelial cells obtained by nasal brushing of CF patients. Following treatment with CF transmembrane conductance regulator (CFTR) modulators, the drugs´ efficacy was confirmed by downregulation of CFTR protein expression, while treatment effectiveness was evaluated by Forskolin-induced swelling functional assays. Given the limited accessibility to patient primary airway epithelial cells, these data strongly support the use of this model for CF pre-clinical studies [43]. In the future, these systems could be adapted to model other types of fibrotic diseases and provide more opportunities for personalised medicine. In another study, a lung organoid model was used to investigate the expression of innate immune receptor, Toll-like receptor 4 (TLR4), and ECM component glycosaminoglycan hyaluronan (HA), are relevant for AECII renewal and repair by limiting fibrosis progression. In vivo, bleomycin treatment of TLR4-/- animals or mice with a HA synthase 2 targeted deletion in AECII were more susceptible to injury when compared with WT mice. For the organoid model, isolated AECII were co-cultured with mouse lung fibroblasts in Matrigel to generate alveolospheres that contain SFTPC positive AECII at the periphery and podoplanin-positive AECI in the interior. Compared to WT alveolospheres, TLR4- and HA-deficient AECII formed fewer and smaller organoids, suggesting that their AECII renewal capacity was compromised. Interestingly, treatment of alveolospheres with exogenous HA increased colony formation efficiency (CFE) in AECII obtained from bleomycin-challenged WT mice. Moreover, 3D cultures of human HTII-280 positive AECII obtained from IPF patients showed lower cell surface HA and reduced CFE compared to AECII isolated from healthy individuals. As with the murine alveolospheres, CFE could be increased by the addition of HA into the media. Ultimately, these results emphasise HAs role in AECII renewal capacity, inferring that future studies will focus on dissecting the molecular mechanisms involved in the loss of HA during pro-fibrotic processes and, if successful, develop strategies for HA restoration on the AECII of IPF patients [44].

Lung organoids are particularly useful to model epithelial-mesenchymal interactions occurring during disease [45,46,47,48,49]. Accordingly, Wilkinson et al. developed a method for the generation of human lung organoids reproducing the distal lung alveolar sac compartment by scaffolding mesenchymal cells into the interstitial space. In this model, alginate beads were coated with polydopamine/collagen 1 and inserted in a bioreactor together with mesenchymal cells allowing an even covering of the epithelial cell types and prompting organoid formation. Notably, mesenchymal cells were indispensable for organoid formation due to their ability to form bead-to-bead bridges. To recreate an IPF environment, TGF-β was added into fetal and iPSC-derived mesenchymal organoid culture. Following TGF-β treatment, highly condensed and smaller organoids grew and exhibited enhanced proliferation markers and *COL1A1* expression with broader α-SMA patches. All these findings were representative of activated myofibroblasts, and subsequent fibroblastic foci formation present in IPF. Importantly, in this study, iPSC-derived mesenchymal cells were successfully transduced with a lentivirus to express mCherry under the control of an ACTA2 promoter that would allow better visualisation, modelling, and analysis of the fibrotic cues occurring during severe fibrosis [50]. In another study, 3D pulmospheres characterised by the presence of multicellular structures embedded in ECM proteins were employed for IPF modelling. Pulmospheres were generated from controls or IPF-derived primary lung cells and contained multiple cell types, including AECII, endothelial cells, macrophages, vascular smooth muscle cells, and myofibroblasts. The presence of an IPF phenotype was confirmed by the increase of α-SMA positive cells radiating outward from the pulmospheres. Interestingly, the extension of invading fibroblasts outside of the organoids´ core provided the authors with a measurement of injury defined as the “zone of invasion” (ZOI) percentage. Following treatment with TGF-β, control lung pulmospheres showed increased ZOI% while treatment of IPF pulmospheres with nintedanib or pirfenidone lowered the invasion area, suggesting that the degree of fibrosis can be modulated using known treatments. Notably, patients with progressive end-stage disease had higher ZOI%, while patients with non-progressive disease but treated with nintedanib or pirfenidone exhibited lower ZOI%, indicating that pulmospheres could reflect, at least to some extent, the in vivo situation [51]. Furthermore, Strikoudis et al. generated a human lung organoid model of fibrosis, employing CRISPR/Cas9 gene-editing technology. In this model, human ESC with engineered mutations in several HPS (Hermansky–Pudlak syndrome) genes caused deformed lysosome-related organelles leading to the development of fibrosis. ESC mutations in HPS1, HPS2, and HPS4 genes generate structurally abnormal organoids characterised by mesenchymal cell aggregation and collagen deposition. Of note, IL-11 was overexpressed in epithelial and mesenchymal cells obtained from HPS1-/- and HPS2-/- organoids. Treatment with IL-11 led to a significant increase of the fibrotic markers, PDGFRα and αSMA, in WT organoids, while deletion of IL-11 in HPS4-/- organoids displayed a morphology similar to WT organoids, implying that prolonged IL-11 exposure may have a significant role in fibrosis induction [52].

Overall, lung organoid systems provide a deeper understanding of physiologically relevant molecular mechanisms of fibrosis that are especially difficult to study in vivo. Importantly, lung organoids derived from patients could be used for high throughput drug screening and prompt the generation of effective and affordable personalised medicine for IPF patients.

### 2.4. Lung in a Chip: Development of Complex Ex Vivo Systems for Pulmonary Fibrosis Modelling

In the last few years, several advances in micro-bioengineering have been achieved to create sophisticated organ-on-a-chip devices containing hollow microchannels lined by living cells. These devices recapitulate certain structural and functional features of tissues and organs, including cell–cell and cell–matrix crosstalk, physical microenvironment, and vascular perfusion [53]. Considering that the human lung is one of the most difficult organs to model in vitro due to its structural and cellular complexity, lung-on-a-chip models are rapidly becoming powerful tools to mimic human lung physiology and pathology [54,55,56,57,58]. For instance, Barkal et al. developed a microscale organotypic model of a human bronchiole to assess fungal and bacterial infection. In this model, pulmonary fibroblasts, primary bronchial epithelial cells, and lung microvascular cells were embedded within a 3D collagen matrix and cultured in a lung-on-chip device with two adjacent lumens to recapitulate the microvascular and airway compartment of the in vivo bronchiole. Notably, this model allows the addition of an interchangeable microbial culture and leucocytes in separate channels that incites an immune response towards the microbial agents via volatile compounds and enables tracking of recruited immune cells [59].

Regarding fibrotic disease modelling, a lung-on-chip model called a “CF-airway-chip” containing primary human CF bronchial epithelial cells, including differentiated ciliated, basal, club, and goblet cells, and pulmonary microvascular endothelial cells, was developed to study CF pathology. Compared to devices containing healthy epithelial cells, the CF-airway-chip recapitulated important features of CF pathology such as increased mucus built-up, cilia density, and beating frequency. After the addition of polymorphonuclear leukocytes (PMNs), circulating PMNs adhered to the endothelium and transmigrated into the airway compartment. In this model, *Pseudomonas aeruginosa* infection led to IL-6, tumour necrosis factor-α, and granulocyte macrophage-colony stimulating factor release in both healthy and CF-airway-chips, while IL-8, a potent PMN chemo-attractant, was significantly higher in CF-airway-chips. Although this model lacks the fibroblast compartment, this model could be a valuable system for the study of pulmonary fibrosis pathophysiology [58]. Another fibrotic model aimed to mimic the human lung upper respiratory airways and model IPF by culturing small airway epithelial cells (SAECs) in one channel and an endothelial-fibroblast-fibrin gel mixture in the outer/central channels. This 3D lung-on-chip design permits the culture of SAECs above a vascular compartment composed of endothelium and fibroblasts. TGF-β treatment of chips containing the healthy or IPF fibroblasts led to upregulation of αSMA expression and downregulation of club and ciliated cell markers, recreating an important component of IPF phenotype. Nevertheless, treatment with the anti-fibrotic agent pirfenidone failed to reduce the fibrotic phenotype in this model, suggesting that further model adaptations are still needed to more closely resemble IPF physiology [60,61].

In a more complex approach, Varone et al. developed a flexible “Open-Top Alveolus-Chip” to mimic the human alveoli by generating a chambered-organotypic epithelium surrounded by two vacuum channels that allow stretch. The top of the chamber contained a removable microchannel capable of being perfused with air or liquid, while the bottom contained a porous flexible membrane permitting transport diffusion. To build vasculature, endothelial cells were cultured in the top chamber and formed a tight monolayer expressing von Willebrand factor, vascular endothelial-cadherin, and platelet and endothelial cell adhesion molecule-1. Epithelial cells (stratified keratinocytes and AEC) could also be cultured in the top chamber. Upon seeding, AECI and II and microvilli were detected, and when cultured with fibroblasts under mechanical stretch, AECII produced surfactant protein C, confirming that exposure to both factors stimulates surfactant production. To study stromal-epithelial crosstalk, the device was challenged with lipopolysaccharide, resulting in upregulation of intracellular adhesion molecule-1 on endothelial cells and production of IL-6, IL-8, and mitochondrial pyruvate carrier-1 on the vascular channel. Notably, this immune response was only detectable when the stromal, vascular, and epithelial cell layers were present. While this device has not been proven to reproduce fibrotic processes, given its complexity and the presence of an alveolar and stromal compartment, it could certainly represent an attractive approach to studying fibrotic diseases [61].

Since lung-on-chip technology is relatively new, further studies are still needed to fully evaluate if such models could be employed to model pulmonary fibrosis. Nonetheless, these devices hold great promise, and it is likely a matter of time until more complex systems can mimic other main features of IPF pathophysiology.

**Table 1 cells-11-01526-t001:** 3D in vivo models employed to study pulmonary fibrosis.

3D Culture System	Species	Source	Cellular Composition	Applicability/ Main Finding	Reference
PA hydrogels	Human	Healthy donors	Lung fibroblasts Distal lung epithelial cells, vascular and mesenchymal compartment	Cellular mechanotransduction	Tse et al. [21]
					Asano et al. [22]
		IPF patients			Chen et al. [30]
	Mouse	Saline/Bleomycin-treated mice		PGE2-COX2 axis modulation in stiff matrixes	Liu et al. [28]
Collagen-rich hydrogels	Human	Healthy donors/ IPF patients		FAK/Akt signalling pathway activation promoting collagen deposition	Giménez et al. [24]
				Eicosanoid biosynthetic enzymes upregulation in fibrotic conditions	Berhan et al. [29]
PCLS	Mouse	Saline/ Bleomycin-treated lung tissue		Drug testing: CSP7	Marudamuthu et al. [33]
				Drug testing: senolytic drugs	Lehmann et al. [36]
				Drug testing: αvβ integrins	Tsoyi et al. [39]
	Human	Healthy donors/ IPF lung tissue		Disease modelling: early fibrosis	Alsafadi et al. [34]
				Drug testing: metformin	Kheirollahi et al. [37]
				Drug testing: SDC2 ligands	Decaris et al. [40]
	Human /Mouse	Healthy lung tissue/Bleomycin-treated lung tissue		Disease modelling: early fibrosis	Lehmann et al. [35]
Lung organoids	Human	CF patients broncho-alveolar lavage	Airway organoids: basal, ciliated, secretory, and club cells	Disease modelling: CF pathophysiology	Sachs et al. [42]
		CF patient nasal brushing	Airway organoids: basal, ciliated, and secretory cells	Personalised medicine: discovery of therapeutic targets and more effective CFTR modulators	Sette et al. [43].
		Healthy donors	Distal alveolar organoids: AECII, AECI, iPSC derived-mesenchymal cells	Disease modelling: epithelial-mesenchymal interactions during fibrosis	Wilkinson et al. [50]
		Healthy/ IPF lung tissue	Pulmospheres: AECII, endothelial cells, macrophages, mesenchymal cells	Disease modelling: myofibroblast activation	Surolia et al. [51]
		Healthy donors	ESC-derived lung organoids–epithelial cells and mesenchymal cells	Disease modelling: mutation in HPS1, HPS2 and HPS4 genes to induce fibrosis	Strikoudis et al. [52]
	Mouse/ Human	HA- and TLR4-deficient mice/ IPF patients	Alveolospheres: AECI, AECII, fibroblasts	Disease modelling: AECII regeneration after fibrosis	Liang et al. [44]
Lung-on-chip	Human	CF patients	CF bronchial epithelial cells and pulmonary microvascular endothelial cells	Disease modelling: CF pathology	Plebani et al. [58]
		Healthy Donors/ IPF patients	SAEC, endothelial cells, fibroblast	Disease modelling: IPF phenotype	Mejías et al. [60]

## 3. Conclusions

As shown above, 3D in vitro lung models have advantages as well as disadvantages that largely depend on the scientific question that needs to be addressed (Figure 1 and Table 2). Regarding IPF, the most used models rely on mouse experimentation, namely, bleomycin-induced pulmonary injury. Although this model mimics certain aspects of IPF pathophysiology, it has been heavily criticized partly due to the presence of inflammation/acute lung injury preceding the fibrotic phase, self-resolution after a few weeks, number of mice needed, strong age- and gender-dependency, among others [62]. Several of these drawbacks can be circumvented or even diminished by the inclusion of 3D in vitro/ex vivo systems. For example, primary human-derived samples such as fibroblasts, iPSC-derived mesenchymal cells, and PCLS can be used after patient consent, therefore, alleviating the ethical concerns of animal experimentation. In addition, these models provide the opportunity to more accurately dissect the cellular and molecular players involved in lung fibrosis, as well as the possibility of high-throughput analyses. One interesting novel and non-invasive approach would be to employ liquid biopsies from IPF patients to facilitate drug testing [63]. The aim of this method would be to evaluate if circulating cells or/and soluble mediators added onto lung-on-chip devices could drive a fibrotic phenotype and then be used to identify specific therapeutic targets.

Ultimately, as these models advance in cellular and structural complexity, their central challenge would be to reveal hidden molecular and cellular signals governing pulmonary fibrosis, which could lead to the discovery and implementation of effective therapies for the treatment and resolution of fibrotic lung diseases.

## Figures and Tables

**Figure 1 cells-11-01526-f001:**
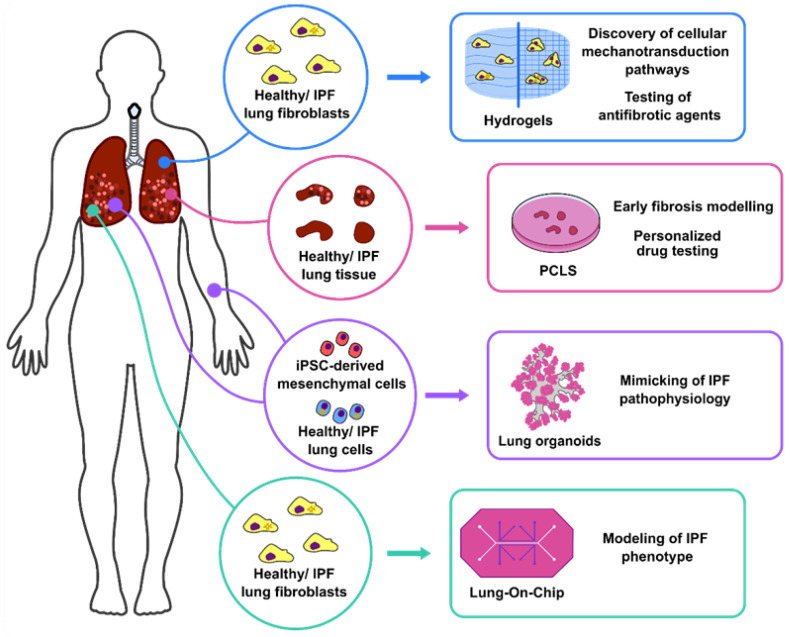
Current human 3D in vitro systems used in IPF research. Lung fibroblasts obtained from healthy donors and IPF patients can be cultured in hydrogels with different stiffness to study cellular mechanotransduction and drug testing. PCLS obtained from patients are being used for modelling early fibrosis and drug screening. In addition, lung organoids derived from lung epithelial stem cells and iPSCs-derived mesenchymal cells have been proven to model IPF pathophysiology, therefore, potentially aiding personalised medicine. Lastly, lung-on-chip devices recapitulate IPF phenotype by culture of fibroblasts isolated from donors and IPF lung tissues together with vascular and lung epithelial cells.

**Table 2 cells-11-01526-t002:** Main strengths and weaknesses of hydrogels, PCLS, lung organoids, and lung-on-chip systems.

3D Culture System	Strengths	Weaknesses
Hydrogels	✓ Tunable matrix stiffness allows mimicking homeostatic and fibrotic microenvironments ✓ Fibroblasts can be obtained from donors and patients	In most instances, not suitable to study intercellular interactions
PCLS	✓ Presence of several lung cell types ✓ Preservation of native lung architecture, microenvironment, and metabolic activity ✓ Patient-derived-PCLS can be used for drug-testing	Cultures are difficult to maintain Analyses involving cell migration are limited
Lung organoids	✓ Progenitor cells from donors and patients can be employed ✓ Suitable to study mesenchymal-epithelial crosstalk ✓ High throughput analyses are possible	Lack vasculature Lack immune cells
Lung-on-chip	✓ Relatively cheap ✓ Mimic lung biochemical microenvironment ✓ Vascular and immune cells can be integrated	Culture and analysis require special equipment Low experimental throughput

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
