# Peer review of "3D In Vitro Models: Novel Insights into Idiopathic Pulmonary Fibrosis Pathophysiology and Drug Screening"

_cells, 2022, doi:10.3390/cells11091526_

Round 1

Reviewer 1 Report

The manuscript entitled:" 3D in vitro models: Novel insights into idiopathic pulmonary fibrosis physiopathology and personalized therapy" focused on a systemic revision of literature data about the role of 3D models in IPF patients is methodologically well written and requires some minor integrations for the publication:

  • In the manuscript, i would suggest to report a conclusion paragrapgh where advantages and weakness of each approach are discusssed. This data may be schematized in a table in order to improve the quality of the manuscript.
  • In my opinion, an other hot topic for IPF patients is liquid biopsy. Could this approach may integrate 3D models? Could the authors describe how this approach may be reliable for IPF patients?

- In my opinion, a figure may be also realized in order to graphically improve the quality of the manuscript.

Author Response

We thank the reviewer for the constructive suggestions. We have addressed these aspects by integrating a new text and a table delineating the advantages and disadvantages of each model in the conclusions section of the revised manuscript. The authors agree that liquid biopsy is an interesting approach, we have included in the revised text an example of how liquid biopsy could be employed in future studies together with a 3D in vitro to improve IPF drug screening. The authors respectfully believe the addition of the pros and cons as a table together with the presented Figure 1 would be sufficient to improve the quality of the manuscript. We sincerely hope that these modifications would fulfill the reviewer´s concerns.

Reviewer 2 Report

This is an interesting review discussing the effectiveness of using 3D in vitro lung models to simulate IPF and develop new treatment targets.  There is little about personalized therapy and does not merit a mention in the title.

Minor point: Are the formats for reference nos 22, 35,36, 51-53 correct?  

Author Response

The authors thank the reviewer for the positive comments and the opportunity to address the remaining concerns. Accordingly, we have now modified the title to better fit the manuscript content and corrected the indicated references’ formats on the revised manuscript.

Reviewer 3 Report

This is a well-written review type of manuscript summarizing recent advances regarding 3D in vitro models as platforms for studying the physiopathology of Idiopathic pulmonary fibrosis (IPF) to get insights into possible personalized therapies. The authors separately illustrated in-depth four types of 3D models where current experimental and therapeutic studies were exampled and discussed, which were supported by relatively updated references.

I do suggest that there be a section where the authors could think of certain apparent research difficulties about IPF and how the four 3D models hypothetically could be integrated and experimentally designed to help solve such difficulties.

Author Response

The authors agree with the reviewer´s suggestion. Therefore, we have included a new paragraph on the revised manuscript summarizing the main difficulties regarding the most common used IPF model, how 3D models can aid to alleviated these issues and an example of how 3D models can be integrated with other approaches to further drive IPF research.